# Epidemiology of Clinical Sporotrichosis in the Americas in the Last Ten Years

**DOI:** 10.3390/jof8060588

**Published:** 2022-05-30

**Authors:** Rigoberto Hernández-Castro, Rodolfo Pinto-Almazán, Roberto Arenas, Carlos Daniel Sánchez-Cárdenas, Víctor Manuel Espinosa-Hernández, Karla Yaeko Sierra-Maeda, Esther Conde-Cuevas, Eder R. Juárez-Durán, Juan Xicohtencatl-Cortes, Erika Margarita Carrillo-Casas, Jimmy Steven-Velásquez, Erick Martínez-Herrera, Carmen Rodríguez-Cerdeira

**Affiliations:** 1Departamento de Ecología de Agentes Patógenos, Hospital General “Dr. Manuel Gea González”, Ciudad de México 14080, Mexico; rigo37@gmail.com (R.H.-C.); yaeesierra@gmail.com (K.Y.S.-M.); 2Sección de Estudios de Posgrado e Investigación, Escuela Superior de Medicina, Instituto Politécnico Nacional, Plan de San Luis y Díaz Mirón, Ciudad de México 11340, Mexico; rodolfopintoalmazan@gmail.com; 3Noncommunicable Diseases Research Group, Universidad La Salle-México, Benjamín Franklin 45, Mexico City 06140, Mexico; 4Sección de Micología, Hospital General “Dr. Manuel Gea González”, Tlalpan, Ciudad de México 14080, Mexico; rarenas98@hotmail.com (R.A.); rodrigo575@hotmail.com (E.R.J.-D.); 5Efficiency, Quality, and Costs in Health Services Research Group (EFISALUD), Galicia Sur Health Research Institute (IIS Galicia Sur), SERGAS-UVIGO, 36213 Vigo, Spain; 6Servicio de Dermatología, Centro Médico Nacional La Raza, Paseo de las Jacarandas S/N, La Raza, Azcapotzalco, Ciudad de México 02990, Mexico; jefegrillo@gmail.com; 7Maestría en Ciencias de la Salud, Escuela Superior de Medicina, Instituto Politécnico Nacional, Plan de San Luis y Díaz Mirón, Ciudad de México 11340, Mexico; victor-espinosa-azul@hotmail.com (V.M.E.-H.); condeesther999@gmail.com (E.C.-C.); 8Laboratorio de Bacteriología Intestinal, Hospital Infantil de México “Dr. Federico Gómez”, Ciudad de México 06720, Mexico; juanxico@yahoo.com; 9Departamento de Biología Molecular e Histocompatibilidad, Hospital General “Dr. Manuel Gea González”, Tlalpan, Ciudad de México 14080, Mexico; ekarri@gmail.com; 10Hospital General San Juan de Dios, 1ra Avenida “A” 10-50, zona 1, Ciudad de Guatemala 01001, Guatemala; js.velasquezb@gmail.com; 11Dermatology Department, Hospital Vithas Vigo and University of Vigo, 36206 Vigo, Spain; 12Campus Universitario, University of Vigo, 36310 Vigo, Spain

**Keywords:** sporotrichosis, *Sporothrix schenckii* sensu stricto, *Sporothrix**schenckii* complex, lymphocutaneous sporotrichosis, fixed cutaneous sporotrichosis, disseminated sporotrichosis, the Americas

## Abstract

Background: Sporotrichosis is a fungal infection caused by species of the *Sporothrix* genus. Presently, the prevalence of sporotrichosis in the Americas is unknown, so this study aims to analyze the cases reported in the past 10 years. Methods: An advanced search was conducted from 2012 to 2022 in English and Spanish in PUBMED, SciELO, and Cochrane, with the terms: “sporotrichosis”, “lymphocutaneous *sporotrichosis*”, “fixed sporotrichosis”, “mycosis”, “*Sporothrix* spp.”, “*Sporothrix* complex”, “*S. schenckii* sensu stricto”, “*S. schenckii* sensu lato”, “*S. globose*”, “*S. brasiliensis*”, “*S. luriei*”. Sporotrichosis is a fungal infection caused by species of the *Sporothrix* genus associated with “pathogenicity” or “epidemiology”. Results: A total of 124 articles were found in the Americas, corresponding to 12,568 patients. Of these, 87.38% of cases were reported in South America, 11.62% in North America, and 1.00% in Central America and the Caribbean. Brazil, Peru, and Mexico had the highest number of cases. The most prevalent etiological agents were *S. schenckii* complex/*Sporothrix* spp. (52.91%), *S. schenckii* (42.38%), others (4.68%), and Not Determined (ND) (0.03%). The most frequent form of the disease was lymphocutaneous infection; however, the infection type was not determined in 5639 cases. Among the diagnostic methods, culture was the most used. Conclusions: There is a high occurrence of cases reported in the literature. South America is the region with the highest number of reports because of its environment (climate, inhalation of spores, etc.), zoonotic transmission (scratches and sneezes from contaminated animals), and possible traumatic inoculation due to outdoor activities (agriculture, gardening, and related occupations). Molecular diagnosis has not been sufficiently developed due to its high cost.

## 1. Introduction

Sporotrichosis is a fungal infection caused by thermo-dimorphic fungi species of the *Sporothrix* genus. Previously, the classification of the species of sporotrichosis was conducted through the classification of the *Sporothrix schenckii* complex, which included *Sporothrix schenckii* sensu stricto, *Sporothrix brasiliensis* (*S. brasiliensis*), *Sporothrix globosa* (*S. globosa*), *Sporothrix luriei* (*S. lurieri*), *Sporothrix pallida* (*S. pallida*), *Sporothrix mexicana* (*S. mexicana),* and *Sporothrix chilensis* (*S. chilensis*) [1,2]. However, since 2016, the taxonomical classification of *Sporothrix* has been changed into a clinical clade that includes *Sporothrix schenckii*, *S.*
*globosa*, *S. brasiliensis, and S. luriei*. On some occasions, the species of the environmental clade, such as *S. pallida*, *S. mexicana,* and *S. chilensis* may cause infection upon contact with an individual [1,2,3,4]. The infections occur mainly cutaneously or subcutaneously with lymphatic involvement [1,2,3,4]. This infection has been considered the most frequent subcutaneous mycosis in Latin America [2]. Such infections can be difficult to diagnose with the naked eye since they can be similar to infiltrative or ulcerative lesions from vascular and inflammatory disorders [1,3].

For this subcutaneous infection to develop, a direct trauma must occur first. For example, inoculation occurs when the skin is punctured by plants with thorns, gardeners are a classic case of this. Also, inoculation can occur through fomites that contact contaminated soil. For instance, people who wear sandals can suffer trauma from stones, firewood, or thorns with fungal spores on their surface [2,3]. With all the above, it can be inferred that this type of fungal infection is associated with regions where the main livelihood is agriculture, that is, in environments where the climate is tropical and subtropical. Another form of transmission, which has been increasing in recent times in some regions of the continent such as Brazil, Argentina, Paraguay, and Panama, has been reported to result from scratches, bites, pecks, and stings from different animals [1,2,3,4].

There are several techniques for detecting sporotrichosis, including Sabouraud dextrose agar cultures, lactophenol blue or erythromycin staining, histopathological studies, and PCR sequencing, among others [5,6,7] (Figure 1).

As for the clinical forms of sporotrichosis, various types have been described, such as the lymphocutaneous, fixed cutaneous, and, as mentioned earlier, the disseminated or hematogenous forms where both organs and tissues can be affected [5,6,7,8,9,10,11,12,13,14,15,16,17,18,19,20,21,22,23,24,25,26,27,28]. The latter is the rarest because the recommended antifungal regimens are usually effective; however, in patients with alterations in cellular immunity, these infections can spread [2,3,4].

## 2. Materials and Methods

An advanced search was conducted in English and Spanish languages in the engines Medical Literature Analysis and Retrieval System Online (MEDLINE/PUBMED), Scientific Electronic Library Online (SciELO), and Cochrane Database. Because the systematic review was performed for the 10 last years (2012–2022), both taxonomical classifications were used. The applied terms were “sporotrichosis”, “lymphocutaneous sporotrichosis”, “fixed sporotrichosis”, “*Sporothrix* spp.” and “*Sporothrix schenckii* complex”, “*S.*
*schenckii* sensu stricto”, and “*S.*
*schenckii* sensu lato”, *Sporothrix schenckii*, (*Sporothrix schenckii*), *Sporothrix brasiliensis* (*S. brasiliensis*), *Sporothrix globosa* (*S. globosa*), *Sporothrix luriei* (*S. lurieri*), *Sporothrix pallida* (*S. pallida*), *Sporothrix mexicana* (*S. mexicana*), and *Sporothrix chilensis* (*S. chilensis*) associated with “pathogenicity” or “epidemiology”. The total number of articles found was 243. The review was performed based on the preferred reporting items for systematic reviews and meta-analyses (PRISMA) (Figure 2). After reading the titles and reviewing the complete text, the most relevant papers to develop this article were identified. At the end of the selection process, 127 articles were chosen. The review was performed based on the preferred reporting items for systematic reviews and meta-analyses (PRISMA).

## 3. Epidemiology of Sporotrichosis in North America

A total of 48 publications related to sporotrichosis were found in North America [5,6,7,8,9,10,11,12,13,14,15,16,17,18,19,20,21,22,23,24,25,26,27,28,29,30,31,32,33,34,35,36,37,38,39,40,41,42,43,44,45,46,47,48,49,50,51,52]. There were 1460 patients in total associated with infection caused by species of the genus *Sporothrix*. According to the previous classification, it was found that in Canada, only two case reports were found, one from Ontario and the other from Toronto [5,6]. In the US, 27 reports containing 1 clinical case were found (81.5% *S. schenckii*, 18.5% *Sporothrix* spp., *S. schenckii* complex, and *S. schenckii sensu lato*) [7,8,9,10,11,12,13,14,15,16,17,18,19,20,21,22,23,24,25,26,27,28,29,30,31,32,33]. Of these, seven cases came from California, three from Oklahoma, two cases from Kansas, Texas, Arizona, Minnesota, and Florida, one case from Michigan, Nebraska, Oregon, Pennsylvania, and finally, one case without a specific city or region. In Mexico, there were 19 reports registered with 1431 reported cases (84.7% *Sporothrix* spp., 14.47% *S. schenckii*, 0.55% *S. globosa*, 0.21% *S. schenckii* sensu stricto, 0.07% *S. mexicana*) [34,35,36,37,38,39,40,41,42,43,44,45,46,47,48,49,50,51,52]. Jalisco reported 1060 cases, Guerrero 150, Nayarit 23, Zacatecas 21, Michoacan 20, Guanajuato 14, Oaxaca 9, Puebla, and San Luis Potosí 8 each, Mexico City 6, Chihuahua, Nuevo León, Querétaro, and Veracruz 2 each, Baja California, Durango, State of Mexico, and Morelos 1 each, and 99 cases were reported with an unspecified city (Table 1). When classifying according to the current taxonomy [1,2,3,4], we can mention that in Canada, 50% of the sporotrichosis was due to *S. schenckii* and 50% to *Sporothrix* spp. [5,6]. In the US, it was reported that *Sporothrix* spp. (85.19%) and *S. schenckii* (14.81%) were responsible for this pathology [7,8,9,10,11,12,13,14,15,16,17,18,19,20,21,22,23,24,25,26,27,28,29,30,31,32,33]. Finally, in Mexico, 85.05% were due to *Sporothrix* spp., 14.33% *S. schenckii*, 0.55% *S. globosa,* and 0.07% *S. mexicana* [34,35,36,37,38,39,40,41,42,43,44,45,46,47,48,49,50,51,52].

The most frequent variety reported was lymphocutaneous with 956 cases, followed by fixed cutaneous with 388 cases, and the disseminated form with 83 [5,6,7,8,9,10,11,12,13,14,15,16,17,18,19,20,21,22,23,24,25,26,27,28,29,30,31,32,33,34,35,36,37,38,39,40,41,42,43,44,45,46,47,48,49,50,51,52]. A lymphocutaneous presentation evolved into a disseminated after 10 months. The least frequent varieties were the disseminated cutaneous with 16 cases, disseminated cutaneous with affected mucous membranes and arthritis with 3 cases each, and the pulmonary form with 2 cases. Finally, lymphadenitis, meningitis, laryngotracheal joint, visceral fungemia, visceral infection associated with fungemia, an atypical presentation, and a visceral presentation with fungemia and mucosal involvement were only reported in one case each. The most common reported etiological agent with the new taxonomical classification was *Sporothrix* spp. with 85.00% (1241/1460), followed by *S. schenckii* with 14.38% (210/1460), *S. globosa* with 0.54% (8/1460), and *S. mexicana* with 0.068% (1/1460) [5,6,7,8,9,10,11,12,13,14,15,16,17,18,19,20,21,22,23,24,25,26,27,28,29,30,31,32,33,34,35,36,37,38,39,40,41,42,43,44,45,46,47,48,49,50,51,52].

In terms of the diagnosis, fungal culture was the most frequently used diagnostic methodology with 33/48, followed by histopathological examination with 20/48. It is worth noting that the histopathological examination was always accompanied by fungal cultures. PCR sequencing was the third method used in 11/48 studies. For this diagnostic tool, the Calmodulin gene was used in 7 cases, the ITS1-2 region in 3 cases, and an unspecified gene in 1 case. Also, the MALDI-TOF and the agglutination latex test were used for diagnosis in two reports. Finally, the use of the Sporotrichin Skin Test and physical examination was mentioned in one report, and one case was reported without describing the employed diagnostic method [5,6,7,8,9,10,11,12,13,14,15,16,17,18,19,20,21,22,23,24,25,26,27,28,29,30,31,32,33,34,35,36,37,38,39,40,41,42,43,44,45,46,47,48,49,50,51,52].

## 4. Epidemiology of Sporotrichosis in Central America and the Caribbean

Only 8 publications with 126 cases of sporotrichosis were found in Central America and the Caribbean [53,54,55,56,57,58,59,60]. In the only article found from Costa Rica during the search period, 57 isolates were analyzed in San José, finding the presence of 2 species: *S. schenckii* sensu stricto (93%), *S. brasiliensis* (3.5%), and *Sporothix* spp. (3.5%) [53]. On the other hand, there were 3 reports in Guatemala with 65 cases (98.5% *Sporothrix* spp. and 1.5% *S. schenckii* sensu stricto), with all cases being from Guatemala City [54,55,56]. Finally, reports of a single case were found in Honduras (Tegucigalpa); the agent responsible for the infection was *S. schenckii,* and in Panama (Correa District), the agent was not determined [57,58]. In the Caribbean, only two reports of *S. schenckii* sensu lato from Cuba were found [59,60]. Regarding the new taxonomic classification, it was determined that in Costa Rica, 93% of the cases were caused by *S. schenckii*, 3.5% by *S. brasiliensis,* and 3.5% by *Sporothix* spp. [53]. Meanwhile, in Guatemala, the main pathogenic agent was *Sporothrix* spp. with 98.5% and *S. schenckii* with 1.5% [54,55,56]. In Honduras and Panama, it was observed that the agent *Sporothrix* spp. was responsible for sporotrichosis, with one case per country (100%) [57,58]. In Cuba, there were two reports of a case due to *Sporothrix* spp., which represents 100% [59,60].

The most frequently reported form was lymphocutaneous with 39 cases (30.95%), followed by fixed cutaneous with 26 (20.63%), the disseminated form with 2 (1.59%), 1 case of chancre (0.79%), and 58 ND cases (46.03%) [53,54,55,56,57,58,59,60]. The most common etiological agents noted were *Sporothrix* spp. with 55.56% (70/126), *S. schenckii* with 42.85% (54/126), and *S. brasiliensis* with 1.59% (2/126) [53,54,55,56,57,58,59,60].

Regarding diagnosis, fungal culture was used as a diagnostic method in all articles (8/8), followed by histopathological examination (5/8). In this case, also, the histopathological examination was always accompanied by fungal cultures. PCR sequencing (2/8) employing the calmodulin gene in one article and the ITS1-2 region in the other was also used as a diagnostic tool. Lastly, diagnosis with microscopy using lactophenol blue was mentioned in two reports (Table 2) [53,54,55,56,57,58,59,60].

## 5. Epidemiology of Sporotrichosis in South America

A total of 68 publications with 11,050 cases of sporotrichosis were found in South America [61,62,63,64,65,66,67,68,69,70,71,72,73,74,75,76,77,78,79,80,81,82,83,84,85,86,87,88,89,90,91,92,93,94,95,96,97,98,99,100,101,102,103,104,105,106,107,108,109,110,111,112,113,114,115,116,117,118,119,120,121,122,123,124,125,126,127,128,129,130,131]. Of these, 4 reports were found in Argentina during the analyzed period with 38 cases, of which 9 were caused by *S. schenckii* sensu stricto (23.68%), 26 by *S. brasiliensis* (68.52%), 1 by *S. globosa* (2.6%), 1 by *S. schenckii* (2.6%), and 1 by *S. schenckii complex* (2.6%) [61,62,63,64]. Brazil reported 42 articles with 5546 analyzed cases [65,66,67,68,69,70,71,72,73,74,75,76,77,78,79,80,81,82,83,84,85,86,87,88,89,90,91,92,93,94,95,96,97,98,99,100,101,102,103,104,105,106], identifying *Sporothrix* spp. and *S. schenckii* complex as the causative agent in 4906 cases (88.46%), *S. schenckii* in 302 (5.45%), *S. brasiliensis* in 125 (2.25%), *Sporothrix* sensu lato in 110 (1.98%), *S. globosa* plus *S. schenckii* in 91 cases (1.64%) *Sporothrix* sensu stricto in 5 (0.09%), *S. globosa* in 4 (0.07%), and *S. mexicana* in 3 (0.05%) during the studied period. In Colombia, 4 reports were found, adding up to 50 cases [56,107,108,109].*S. Schenckii* sensu stricto was identified in 22 cases (44.00%), *Sporothrix* spp. in 15 (30.00%), *S. globosa* in 12 (24.00%) and S. *schenckii* sensu lato in 1 (2.00%). Likewise, in Chile, 3 reported cases detected *Sporothrix* spp. in 1 (33.33%), *S. globosa* in 1 (33,33%), and *Sporothrix pallida* in 1 (33.33%) [110,111,112]. A total of 13 cases of *Sporothrix* spp. and *S. schenckii* complex (100%) were reported in Paraguay [113,114]. In Peru, from 4792 cases, *S. schenckii* was found in 4656 (97.16%), *Sporothrix* spp. and the *Sporothrix* complex in 116 (2.42%), *S. schenkii* sensu stricto in 19 (0.40%), and *Sporothrix* sensu lato in 1 (0.02%) [115,116,117,118,119,120,121,122,123]. There was 1 report of 157 cases of *Sporothrix* spp. (100%) found in Uruguay [124]. Finally, there were 4 reports from Venezuela with 452 cases of *Sporothrix* spp., and the *Sporothrix* complex was found in 220 of those cases (48.67%), *S. schenckii* sensu lato in 130 (28.76%), *S. schenckii* in 42 (9.29%), *S. schenckii* sensu stricto in 17 (3.76%), *S. globosa* in 39 (8.63%), 1 case of *Ophiostoma stenoceras* (0.22%) and 3 cases were ND (0.66%) [125,126,127,128].

Regarding the new taxonomic classification, in Argentina, 26.31% were *S. schenckii*, 68.42% *S. brasiliensis*, 2.63% *S. globosa,* and 2.63% *Sporothrix* spp. [61,62,63,64]. In Brazil, the main pathogenic agent was *Sporothrix* spp. with 95.56%, *S. brasiliensis* 2.25%, *S. globosa* plus *S. schenckii* 1.64%, *S. schenckii* 0.41%, *S. globosa* 0.07%, and *S. mexicana* 0.05% [65,66,67,68,69,70,71,72,73,74,75,76,77,78,79,80,81,82,83,84,85,86,87,88,89,90,91,92,93,94,95,96,97,98,99,100,101,102,103,104,105,106]. In Colombia, *S. schenckii* 44.00%, *Sporothrix* spp. 32.00%, and *S. globosa* 24.00% were the principal mycotic agents [56,107,108,109]. Regarding Chile, the pathogenic agents were *Sporothrix* spp., *S. globose*, and *S. pallida* (33.33% each) [110,111,112]. In Paraguay, the unique agent found was *Sporothrix* spp. (100%) [113,114]. For Peru, the most important pathogenic agents were *Sporothrix* spp. (99.54%), and *S. schenckii* (0.46%) [115,116,117,118,119,120,121,122,123]. In Uruguay, 100% of the cases were due to *Sporothrix* spp. (100%) [124]. In Venezuela, *Sporothrix* spp. (80.04%), *S. schenckii* (13.38%), and *S. globose* (6.57%) were the types of *Sporothrix* agents [125,126,127,128].

The most frequent types of disease were lymphocutaneous with 3293 cases (29.47%), fixed cutaneous with 1947 (17.43%), disseminated cutaneous with 34 (0.30%), systemic form with 18 (0.16%), and others with 177 cases (1.60%). However, there were 5702 cases (51.04%) with undetermined types from all the cases diagnosed as sporotrichosis [56,61,62,63,64,65,66,67,68,69,70,71,72,73,74,75,76,77,78,79,80,81,82,83,84,85,86,87,88,89,90,91,92,93,94,95,96,97,98,99,100,101,102,103,104,105,106,107,108,109,110,111,112,113,114,115,116,117,118,119,120,121,122,123,124,125,126,127,128].

The most common reported etiological agent with the new taxonomical classification was *Sporothrix* spp. with 95.12% (10,511/11,050), followed by *S. schenckii* with 1.23% (136/11,050), *S. brasiliensis* with 2.27% (251/11,050), *S. globosa* plus *S. schenckii* with 0.82% (91/11,050), *S. globosa* with 0.52% (57/11,050), *S. mexicana* 0.027% (3/11,050), and *S. pallida* with 0.009% (1/11,050) [56,61,62,63,64,65,66,67,68,69,70,71,72,73,74,75,76,77,78,79,80,81,82,83,84,85,86,87,88,89,90,91,92,93,94,95,96,97,98,99,100,101,102,103,104,105,106,107,108,109,110,111,112,113,114,115,116,117,118,119,120].

With reference to diagnosis, fungal culture was used as a diagnostic methodology in almost all articles (67/71), followed by PCR sequencing (26/71), where the calmodulin gene (15/23), the ITS 1-2 region (6/23), and other genes (15/23) were used. Other types of diagnoses (12/71) were applied, such as direct microscopy (19/71), histopathological examination, always accompanied by fungal culture (18/71), and microscopy with lactophenol blue (9/71) (Table 3).

## 6. Discussion

A total of 124 publications were found with reports related to sporotrichosis in the Americas in the last 10 years, with 12,636 patients associated with infection caused by species of the genus *Sporothrix*. Interestingly, it was observed that 87.45% (11,050) of these cases were reported in South America, 11.55% (1460) in North America, and 1.00% (126) in Central America and the Caribbean [5,6,7,8,9,10,11,12,13,14,15,16,17,18,19,20,21,22,23,24,25,26,27,28,29,30,31,32,33,34,35,36,37,38,39,40,41,42,43,44,45,46,47,48,49,50,51,52,53,54,55,56,57,58,59,60,61,62,63,64,65,66,67,68,69,70,71,72,73,74,75,76,77,78,79,80,81,82,83,84,85,86,87,88,89,90,91,92,93,94,95,96,97,98,99,100,101,102,103,104,105,106,107,108,109,110,111,112,113,114,115,116,117,118,119,120,121,122,123,124,125,126,127,128]. The countries that presented the highest number of cases during the analyzed period were Brazil (5546—43.89%), Peru (4792—37.92%), and Mexico (1431—11.32%). It should be noted that in the case of Brazil and Peru, there were various reports with several cases from a time period ranging from 25 to 50 years [66,67,75,86]. As previously mentioned, sporotrichosis is a disease caused by a thermodymorphic fungus of the genus *Sporothrix*. It is known that this subcutaneous disease, although cosmopolitan, generally occurs in both tropical and subtropical regions. The latter could explain, in some part, the high prevalence in Latin America, being endemic in this region [1,2,3,4,129]. However, three countries (Brazil, Peru, and Mexico) have specific characteristics that increase the number of cases. In Brazil and adjacent countries (for example, Argentina and Paraguay), an increasing number of cases have been associated with zoonotic infection, mainly from infected cats through scratches or sneezes [3,4]. Since the zoonotic transmission of *S. brasiliensis* is the most important form of communication, it is recommended that hygienic measures be taken regarding domestic animals such as cats, rodents, etc., due to possible infections. If it is diagnosed in animals, it must be treated immediately, and gloves must be used when handling animals with injuries [2,3,4].

In Mexico, sporotrichosis is considered endemic and an occupational disease due to the different sources of infection. The climate of some regions in Mexico is perfect for the characteristics of this type of mycosis to increase its incidence. Although tropical and subtropical climates are preferred by this fungus, in this country, the cold and dry seasons are the contagion peaks of these pathological agents. The states that are more affected are Mexico City, Puebla, Jalisco, Michoacan, the State of Mexico, and Guanajuato. In these states, agriculture is one of the most important economic activities, which explains the high incidence of the *Sporothrix* contagion [130]. Thus, the principal recommendation in this region is the use of gloves or long-sleeved clothing when carrying out work activities where these species are endemic.

Regarding the etiological agents responsible for the types of sporotrichosis, it is important to specify that they were referred to both in the table and in the text in the way they were named in the articles that were analyzed. Since most of them were written and published before the changes in taxonomical classification, they do not consider the clinical and environmental clades classification instead of the *Sporothrix schenckii* complex.

As for the etiological agent of sporotrichosis, the most prevalent, according to the reports with the old taxonomical classification, were *S. schenckii* complex and *Sporothrix* spp. with 6624 cases (52.41%), *S. schenckii* with 5302 (41.95%), *S. schenckii* sensu lato with 245 (1.94%), *S. schenckii* sensu stricto with 147 (1.16%), *S. brasiliensis* with 153 (1.21%), *S. globosa* plus S*. Schenckii* sensu stricto with 91 (0.72%), *S. globosa* with 65 (0.51%), *S. mexicana* with 4 (0.03%), *S. pallida* 1 (0.008%), *Ophiostoma stenoceras* 1 (0.008%), and 4 ND cases (0.032%). It is worth mentioning that, although there are other species, such as *Sporothrix luriei*, there were no reports found in the studied period in the Americas [1,131].

Likewise, within the systematic review, *Ophiostoma stenoceras* appears, which in the taxonomic classification of *Sporothrix* is represented in its sexual state, in the year the report was made. Nevertheless, in 2016, Beer et al. concluded through phylogenetic analyzes that the genus *Sporothrix* was different from the genus *Ophiostoma*, but that was before considering its sexual state. Officially, the sexual status of *Sporothrix* is not known, and in this case, *Ophiostoma stenoceras* was included according to the regulations that governed the taxonomy before the divorce between the two genders occurred [132].

After analyzing the articles to carry out the classification according to the new taxonomy, we found that the most common reported etiological agent was *Sporothrix* spp. with 94.34% (11,922/12,636), followed by *S. schenckii* with 3.16% (400/12,636), *S. brasiliensis* with 1.21% (153/12,636), *S. globosa* plus *S. schenckii* with 0.72% (91/12,636), *S. globosa* with 0.51% (65/12,636), *S. mexicana* 0.03% (4/12,636), and *S. pallida* with 0.007% (1/12,636) [61,62,63,64,65,66,67,68,69,70,71,72,73,74,75,76,77,78,79,80,81,82,83,84,85,86,87,88,89,90,91,92,93,94,95,96,97,98,99,100,101,102,103,104,105,106,107,108,109,110,111,112,113,114,115,116,117,118,119,120,121,122,123,124,125,126,127,128].

On the other hand, the most frequent type of sporotrichosis was the lymphocutaneous with 4288 cases, followed by the fixed cutaneous with 2340 cases, the disseminated or systemic with 103, the disseminated cutaneous with 52 cases, other with 215 cases, and 5760 cases were ND. By being a subcutaneous mycosis, the lymphocutaneous form is the most frequent one because sporotrichosis mainly affects the lymph nodes of the skin and the subcutaneous tissue, producing ulcers and thereby affecting the lymphocutaneous system [133]. The infection begins in the form of an inoculation chancre. Subsequently, erythematous nodular lesions arise, which follow the trajectory of the lymphatic vessels, mainly affecting the face and upper and lower limbs. Another common form is the fixed cutaneous, which occupies the second place in the Americas to the present date. This type is of a fixed form at the inoculation site of the fungus, affecting mainly children, and it is observed as a verrucous plaque. Its presence demonstrates a high immunity response from the patient. Being a disease of this body region, it has a low prevalence in other organs or tissues. However, the disseminated or hematogenous form may be cutaneous or systemic [5,6,7,8]. Systemic sporotrichosis can cause respiratory and lung disorders, osteomyelitis, arthritis, and meningitis. It is important to note that the type of condition affecting the patients (5760 cases) was not mentioned in several of the reported cases analyzed in this publication.

Concerning diagnosis, various methods, both phenotypic and genotypic, have been used to detect the infection caused by this etiological agent [129]. Within the phenotypic methods, we can name (1) mycological cultures. This technique seeks the growth of the colony in a radial form (approximately 3 to 4 days) with a creamy consistency, and subsequently, the development of mycelium is observed for its identification (Gold Standard). Finally, it is suggested to perform a lactophenol blue staining to observe the microconidia in a sympodial arrangement along the mycelium. (2) serological diagnosis using sporotricine and immunodiffusion tests, immunoelectrophoresis, latex agglutination, etc. (3) histopathological diagnosis, an excisional biopsy of the nodular lesion is performed that may show granulomatous and necrotizing dermatitis, which can be stained with Hematoxylin and Eosin (HE) Schiff’s Periodic Acid (PAS), or Grocott-Gomori Methenamine Silver (MSG) to confirm the presence of asteroid bodies [5,6,7,8].

Nonetheless, genotypic identification tests are preferred since phenotypic techniques have disadvantages, such as being laborious, presenting variable results from the clinical field, and requiring many samples to reach a diagnosis. Therefore, different PCR techniques have been used for genotypic identification tests utilizing diverse genetic or molecular markers that have been developed [5].

In this systematic review, the culture turned out to be the most used diagnostic method throughout the continent, being performed in 107 of the 127 articles reviewed. Histopathological examination was the second most used diagnostic technique, found in 43 publications. In addition, PCR sequencing was used 38 times, direct microscopy 21 times, and microscopy with lactophenol blue was reported in 9 articles. Lastly, other techniques were used to detect sporotrichosis; however, these were not utilized routinely.

## Figures and Tables

**Figure 1 jof-08-00588-f001:**
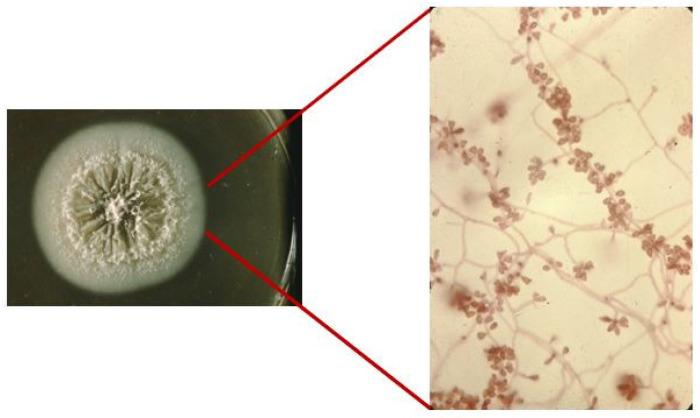
*Sporothrix* spp. culture and erythromycin staining 40×.

**Figure 2 jof-08-00588-f002:**
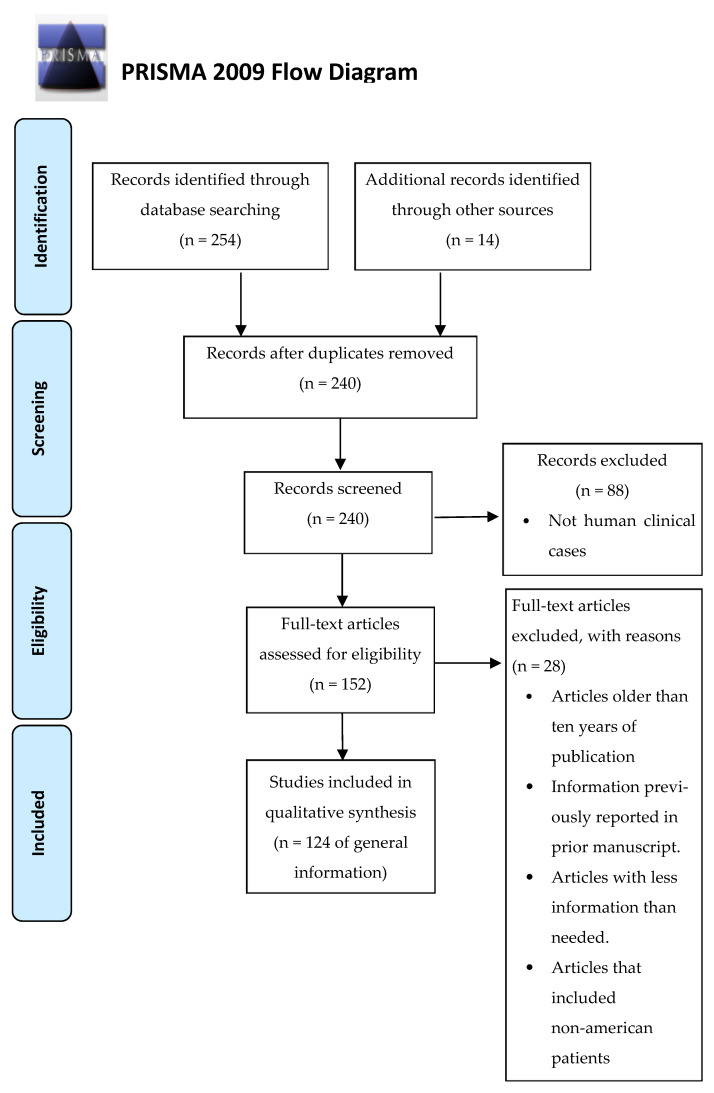
Flowchart of the different phases of the systematic review.

**Table 1 jof-08-00588-t001:** Epidemiology of Sporotrichosis in North America.

Region	Country	City	Number of Reported Cases	Vulnerable Population		Diagnostic Method	Type of Sporotrichosis	Etiological Agents (%)		References
				Sex	Age (Years)			Taxonomy		
								Before 2017	After 2017	
North America	Canada	Ontario	1	Male	44	PCR sequencing (ITS region)	Disseminated	*S. schenckii*	*S. schenckii*	[5]
		Toronto	1	Male	78	Fungal culture, Biopsy (Histopathology)	Lymphocutaneous	*S. schenckii* complex	*Sporothrix* spp.	[6]
	USA	California	1	Female	7	Fungal culture Biopsy (Histopathology)	Lymphocutaneous	*S. schenckii*	*Sporothrix* spp.	[7]
		Minnesota	1	Male	61	Fungal culture	Disseminated	*S. schenckii*	*Sporothrix* spp.	[8]
		ND	1	Female	87	Fungal culture	Lymphocutaneous on the eyelid	*S. schenckii*	*Sporothrix* spp.	[9]
		Pennsylvania	1	Male	67	Fungal culture Biopsy (Histopathology)	Lymphocutaneous	*S. schenckii*	*Sporothrix* spp.	[10]
		Texas	1	Male	34	Fungal culture Biopsy (Histopathology)	Disseminated	*Sporothrix* spp.	*Sporothrix* spp.	[11]
		Texas	1	Male	9	Fungal culture Biopsy (Histopathology)	Lymphocutaneous on the eyelid	*S. schenckii*	*Sporothrix* spp.	[12]
		California	1	Female	41	Fungal culture	Lymphocutaneous	*S. schenckii*	*Sporothrix* spp.	[13]
		Oregon	1	Male	53	Fungal culture	Disseminated	*Sporothrix* spp.	*Sporothrix* spp.	[14]
		Oklahoma	1	Male	66	Latex agglutination test	Disseminated	*S. schenckii*	*Sporothrix* spp.	[15]
		Florida	1	Male	33 month-Old	Fungal culture Biopsy (Histopathology)	Atypical lymphadenitis	*S. schenckii*	*Sporothrix* spp.	[16]
		Minnesota	1	Male	49	Fungal culture	Pulmonary sporotrichosis	*Sporothrix* spp.	*Sporothrix* spp.	[17]
		Arizona	1	Male	56	Fungal culture	Lymphocutaneous and disseminated (10 months later)	*S. schenckii*	*Sporothrix* spp.	[18]
		California	1	Male	39	Fungal culture	Sporothrical arthritis	*S. schenckii*	*Sporothrix* spp.	[19]
		California	1	Male	89	Fungal culture Biopsy (Histopathology)	Disseminated	*S. schenckii*	*Sporothrix* spp.	[20]
		Michigan	1	Female	57	Fungal culture Biopsy (Histopathology)	Lymphocutaneous	*S. schenckii*	*Sporothrix* spp.	[21]
		California	1	Male	34	Latex agglutination test	Chronic meningitis	*S. schenckii*	*Sporothrix* spp.	[22]
		Kansas	1	Male	33	Fungal culture MALDI-TOF	Sporothrical arthritis	*S. schenckii*	*Sporothrix schenckii*	[23]
		Oklahoma	1	Male	44	Fungal culture Biopsy (Histopathology)	Pulmonary sporotrichosis	*S. schenckii* sensu lato	*Sporothrix* spp.	[24]
		California	1	Male	41	Fungal culture	Sporothrical arthritis	*S. schenckii*	*Sporothrix* spp.	[25]
		California	1	Female	35	Fungal culture	Disseminated	*S. schenckii*	*Sporothrix* spp.	[26]
		Nebraska	1	Male	62	Fungal culture Biopsy (Histopathology)	Disseminated	*S. schenckii*	*Sporothrix* spp.	[27]
		Boston	1	Female	35	MALDI-TOF	Fixed cutaneous	*S. schenckii*	*S. schenckii*	[28]
		Kansas	1	Male	30	Fungal culture Biopsy (Histopathology)	Disseminated	*S. schenckii*	*Sporothrix* spp.	[29]
		Florida	1	Male	76	History and physical examination	Lymphocutaneous	*Sporothrix* spp.	*Sporothrix* spp.	[30]
		Oklahoma	1	Male	23	Fungal culture	Lymphocutaneous	*S. schenckii* complex	*Sporothrix* spp.	[31]
		Washington	1	Female	44	Fungal culture PCR sequencing (ITS 1–2)	Disseminated	*S. schenckii*	*S. schenckii*	[32]
		Arizona	1	Female	72	PCR DNA sequencing	Laryngotracheal granulomatous disease	*S. schenckii*	*S. schenckii*	[33]
	Mexico	Veracruz	1	Male	39	Fungal culture Biopsy (Histopathology)	Atypical	*S. schenckii*	*Sporothrix* spp.	[34]
		Puebla	1	Male	36	Fungal culture Biopsy (Histopathology)	Disseminated	*S. schenckii*	*Sporothrix* spp.	[35]
		Oaxaca	1	Male	13	Fungal culture	Lymphocutaneous on the left hand, forearm, and upper arm	*Sporothrix* spp.	*Sporothrix* spp.	[36]
		Mexico City	1	Male	54	Fungal culture Biopsy (Histopathology)	Disseminated (Testicular involvement)	*S. schenckii*	*Sporothrix* spp.	[37]
		Guerrero	1	Female	36	Fungal culture Biopsy (Histopathology)	Disseminated	*Sporothrix* spp.	*Sporothrix* spp.	[38]
		Durango	1	Male	68	Fungal culture Biopsy (Histopathology)	Disseminated	*Sporothrix* spp.	*Sporothrix* spp.	[39]
		ND	24	Male (16) Female (8)	Average: 35.5	PCR sequencing (calmodulin gene)	Cutaneous disseminated 16 (66.7%) Cutaneous disseminated + Mucosal 3 (12.5%) Joint 1 (4.1%) Visceral 1 (4.1%) Fungaemia 1 (4.1%) Mucosal + Visceral + Fungemia: 1 (4.1%) Visceral + Fungaemia 1 (4.1%)	*S. schenckii* 23 (95.5%). *S. globosa* 1 (4.5%)	*S. schenckii* 23 (95.5%). *S. globosa* 1 (4.5%)	[40]
		ND	55	Male (34) ND Female (18)		Sporotrichin Skin Test Fungal culture	Lymphocutaneous 32 (58.2%) Fixed cutaneous 19 (34.5%) Disseminated 4 (7.3%)	*S. schenckii* 54 (98%) *S. globosa* 1 (2%)	*S. schenckii* 54 (98%) *S. globosa* 1 (2%)	[41]
		Guerrero	73	Male (33) Female (40)	Average: 25.8	Fungal culture Biopsy (Histopathology)	Lymphocutaneous: 41 (56.16%) Fixed cutaneous 24 (32.87%) Disseminated 8 (10.95%)	*S. schenckii*	*S. schenckii*	[42]
		Chihuahua	1	Female	84	Multiplex PCR (Calmodulin gene)	Fixed cutaneous (Auricular sporotrichosis)	*S. schenckii* (sensu stricto)	*S. schenckii*	[43]
		Baja California	1	Male	23	Fungal culture Biopsy (Histopathology)	Lymphocutaneous	*S. schenckii*	*Sporothrix* spp.	[44]
		San Luis Potosi 8 Puebla 3 Mexico City 2 Queretaro 2 Guanajuato 2 Jalisco 1 Zacatecas 1 Michoacan 1 Morelos 1 State of Mexico 1	22	ND		PCR sequencing (Calmodulin and *calcium-calmodulin-dependent kinase* genes)	Lymphocutaneous: 17 (77.3%) Fixed cutaneous 4 (18.2%) Disseminated 1 (4.5%)	*S. schenckii*: 18 (81.8%) *S. globosa* 4 (18.2%)	*S. schenckii*: 18 (81.8%) *S. globosa* 4 (18.2%)	[45]
		Puebla 4 Nuevo Leon 2 Oaxaca 6 Mexico City 3 Jalisco 2	17	ND		PCR sequencing (Calmodulin gene)	Lymphocutaneous: 16 (94.11%) Disseminated: 1 (5.88%)	*S. schenckii*: 16 (94.11%) *S. globosa* 1 (5.88%)	*S. schenckii*: 16 (94.11%) *S. globosa* 1 (5.88%)	[46]
		Guerrero	76	Male (35) Female (41)	<18: 37 >18: 39	Fungal culture Biopsy (Histopathology)	Lymphocutaneous 43 (56.8%) Fixed cutaneous 24 (32.3%) Disseminated 8 (11%)	*Sporothrix* spp.	*Sporothrix* spp.	[47]
		Jalisco 1057 Nayarit 23 Zacatecas 20 Michoacan 19 Guanajuato 12 Veracruz 1 Chihuahua 1	1134	Male (669) Female (465)		ND	Lymphocutaneous: 782 (69%) Fixed cutaneous: 308 (27.2%) Disseminated 44 (38.8%)	*S. schenckii* complex	*Sporothrix* spp.	[48]
		ND	1	Male	45	PCR sequencing (Calmodulin gene)	Disseminated	*S. schenckii* complex	*S. schenckii*	[49]
		ND	1	Male	56	Fungal Culture Biopsy (Histopathology) PCR sequencing (ITS and calmodulin gene)	Fixed cutaneous sporotrichosis	*S. mexicana*	*S. mexicana*	[50]
		ND	18	Male (10) Female (8)	ND	PCR sequencing (ITS regions)	Lymphocutaneous 13 (72.2%) Fixed cutaneous 5 (27.8%)	*S. schenckii* 17 (94.4%) *S. globosa:* 1 (5.6%)	*S. schenckii* 17 (94.4%) *S. globosa:* 1 (5.6%)	[51]
		Oaxaca	2	Male	61	Multiplex PCR (Calmodulin gene)	Fixed cutaneous 1 (50%) Disseminated 1 (50%)	*S. schenckii* sensu stricto	*S. schenckii*	[52]
				Male	21					

ND: Not Determined.

**Table 2 jof-08-00588-t002:** Epidemiology of Sporotrichosis in Central America and the Caribbean.

Region	Country	City	Number of Reported Cases	Vulnerable Population		Diagnostic Method	Type of Sporotrichosis	Etiological Agents (%)		References
				Sex	Age (Years)			Taxonomy		
								Before 2017	After 2017	
Central America	Costa Rica	San José	57 (1994–2015)	No data		Direct microscopy, culture, PCR (enzymatic restriction and sequencing of the calmodulin gen)	ND	*S. schenckii* sensu stricto 53 (93%) *S. brasiliensis* 2 (3.5%) *Sporothrix spp.* 2 (3.5%)	*S. schenckii* 53 (93%) *S. brasiliensis* 2 (3.5%) *Sporothrix spp.* 2 (3.5%)	[53]
	Guatemala	Guatemala City	11	Male 7 Female 4	Average 49 years	Fungal culture, Histopathology	Fixed cutaneous 9 (81.8%) Lymphocutaneous 2 (18.2%)	*Sporothrix* spp. (100%)	*Sporothrix* spp. (100%)	[54]
		Guatemala City	53 (2007–2016)	Male 33 Female 20	Average 44.1 years	Fungal culture, microscope with Lactophenol cotton blue	Lymphocutaneous 33 (62.2%) Fixed cutaneous 17 (32.1%) Disseminated 2 (3.8%) Chancre 1 (1.9%)	*Sporothrix schenckii* complex. (100%)	*Sporothrix* spp. (100%)	[55]
		Guatemala City	1	ND		Fungal culture, PCR sequencing (ITS 1- 2 and β -tubulin)	ND	*Sporothrix schenckii* sensu stricto	*Sporothrix schenckii*	[56]
	Honduras	Tegucigalpa	1	Male 1	14 years	Fungal culture	Lymphocutaneous 1 (100%)	*S. schenckii*	*Sporothrix* spp.	[57]
	Panamá	Chorrera District	1	Male 1	34 years	Clinical, Direct Microscopy, Fungal culture.	Lymphocutaneous 1 (100%)	ND	*Sporothrix* spp.	[58]
Caribbean	Cuba	Pinar del Río	1	Female 1	57 years	Histopathology Fungal culture	Lymphocutaneous	*Sporothrix schenckii* sensu lato (100%)	*Sporothrix* spp. (100%)	[59]
		Cumanayagüa	1	Male	67	Histopathology, Fungal culture, Microscopy with lactophenol cotton blue	Lymphocutaneous	*Sporothrix schenckii* sensu lato (100%)	*Sporothrix* spp. (100%)	[60]

ND: Not Determined.

**Table 3 jof-08-00588-t003:** Epidemiology of Sporotrichosis in South America.

Region	Country	City	Number of Reported Cases	Vulnerable Population		Diagnostic Method	Type of Sporotrichosis	Etiological Agents (%)		References
				Sex (Number of Cases)	Age (Years)			Taxonomy		
								Before 2017	After 2017	
South America	Argentina	Provincia de Chaco	1	Female	65	Bronchoalveolar lavage (BAL), Giemsa stain Fungal culture PCR sequencing (ITS 1–2)	Pulmonary	*S. schenckii*	*S. schenckii*	[61]
		Buenos Aires	16	Male (4) Female (10) ND (7)	Average 32.5	Fungal culture and PCR sequencing (Calmodulin gene)	Lymphocutaneous 33 (42.9%) Fixed cutaneous 17 (19.0%) ND (38.1%)	*S. brasiliensis*	*S. brasiliensis*	[62]
		Misiones	1			Fungal culture and PCR sequencing (Calmodulin gene)				
		El Calafate	4			Fungal culture, PCR sequencing (Calmodulin gene) and histopathology				
		Buenos Aires	15	ND		Fungal culture (agar potato dextrose and brain heart infusion agar) PCR sequencing (Calmodulin gene)	ND	*S. schenckii* sensu stricto 9 (56.5%) *S. brasiliensis* 5 (34.7%) *S. globosa* 1 (8.7%)	*S. schenckii* 9 (56.5%) *S. brasiliensis* 5 (34.7%) *S. globosa* 1 (8.7%)	[63]
		Buenos Aires	1	Female	5	Direct microscopy, Fungal culture (Sabouraud agar), Histopalology	Lymphocutaneous	*S. schenckii complex*	*Sporothrix* spp.	[64]
	Brazil	Rio de Janeiro (Duque de Caxias)	827 from 2007–2016	Female (541) Male (286)	42	Fungal culture	ND	*Sporothrix* spp.	*Sporothrix* spp.	[65]
		Rio de Janeiro ND Teresópolis ND	1563 (1999–2008 = 50 (3.20%))	Male (16) Female (34)	Average 47	Direct microscopy, Fungal culture, PCR sequencing (calmodulin gene)	Lymphocutaneous 24 (48%) Fixed cutaneous 15 (30%) Disseminated cutaneous 6 (12%) disseminated (involving internal tissues) 5 (10%)	*S. brasiliensis* 45 45 (90%) *S. schenckii* sensu stricto 5 (10%)	*S. brasiliensis* 45 (90%) *S. schenckii* 5 (10%)	[66]
		Rio de Janeiro	Group 1 48 (1.33%) Group 2 3570 (98.67%) 1987–2013	Group 1 HIV patients Male 33 Female 15 Group 2 Immunocompetent patients Male (1102) Female (2468)	Average: 38.4 Averag: 46.3	Direct microscopy, Fungal culture.	ND	*Sporothrix* spp.	*Sporothrix* spp.	[67]
		Rio de Janeiro	21/1750 cases in HIV patients (1.2%) from 1999–2009	Male (16) Female (5)	Average: 41.2	Direct microscopy, Fungal culture, Histopathology	Lymphocutaneous 7 (33.3%) Disseminated 7 (33.3%) widespread cutaneous 5 (23.8%) fixed cutaneous 2 (9.5%)	*S. schenckii* sensu lato	*Sporothrix* spp.	[68]
		Rio de Janeiro 16 Duque de Caxias 6 São João de Meriti 2 São Gonçalo 1 Maricá 1	26 from 2007–2017	Female (19) Male (7)	Average: 25	Direct microscopic, Fungal culture	Primary ocular 21 (80.8%) Associated cutaneous disease (3 lymphocutaneous, 1 the fixed cutaneous and 1 the disseminated 5 (19.2%)	*Sporothrix* spp.	*Sporothrix* spp.	[69]
		Rio de Janeiro	86 from 2009–2017	Male (26) Female (60)	Average: 36.3 Average: 46	Fungal culture Histopathology	ND	*Sporothrix* spp.	*Sporothrix* spp.	[70]
		Espíritu Santo	73 from 2016–2019	Male Female	ND	Fungal culture, Microscopy with lactophenol cotton blue, PCR sequencing (Calmodulin gene and *Mating type* (MAT) gene)	ND	*S. brasiliensis* 55 (76%) *S. schenckii* sensu stricto 18 (24%)	*S. brasiliensis* 55 (76%) *S. schenckii* 18 (24%)	[71]
		Espíritu Santo	171 cases from 1982–2012	Male (138) Female (33)	Average: 33.42	Fungal culture	ND	*Sporothrix* spp.	*Sporothrix* spp.	[72]
		Rio Grande do Sul	83 from 2010–2016	ND		Fungal culture	ND	*Sporothrix* spp.	*Sporothrix* spp.	[73]
		Rio Grande do Sul	43 from 2006–2015	Male (31) Female (7)	Average: 43	Fungal culture	Lymphocutaneous 22 (51%) Fixed cutaneous 14 (32.5%) Disseminated cutaneous 1 (2.5%) ND 6 (14%)	*Sporothrix* spp.	*Sporothrix* spp.	[74]
		Minas Gerais	282	Male (153) Female (129)	Average: 42.52	Fungal culture, Sporotrichin test, Histophatology, Production of *S. schenckii* antigens, Enzyme-linked immunosorbent assay	ND	*S. schenckii*	*Sporothrix* spp.	[75]
		Brasilia	91 from 1993–2018	Male (64) Female (27)	ND	Direct microscopy, Fungal culture, PCR sequencing (Calmodulin gene)	Lymphocutaneous 34 (37.36%) Cutaneous fixed 6 (6.59%) Disseminated 5 (5.49%) ND 46 (50.55%)	*S. globosa* (ND) *S. schenckii* (ND)	*S. globosa* (ND) *S. schenckii* (ND)	[76]
		São Paulo	25 from 2003–2013.	Male (18) Female (7)	Average: 42.48	Fungal culture Histopathology	Lymphocutaneous 20 (80%) Fixed cutaneous 5 (20%)	*S. schencki* sensu lato	*Sporothrix* spp.	[77]
		São Paulo	20 from 2012–2020	Male (9) Female (11)	Average: 2.2	Direct microscopy, Fungal culture, Histopathology	Lymphocutaneous 10 (50%) Multiple-inoculation 5 (25%) Fixed-cutaneous 3 (15%) Ocular-mucosal 2 (10%)	*Sporothrix* spp.	*Sporothrix* spp.	[78]
		Rio de Janeiro	1	Male	35	Direct microscopy, fungal culture	Osteomyelitis	*S. schenckii* complex	*Sporothrix* spp.	[79]
		Rio de Janeiro	1	Female	68	Direct microscopy (KOH), fungal culture (Sabouraud Dextrose Agar 2%, and Mycosel Agar, Brain Heart Infusion Agar, Potato Dextrose Agar), Lactophenol Cotton Blue and MALDI-TOF MS	Ocular	*S. brasiliensis*	*S. brasiliensis*	[80]
		Rio Grande do Norte	1	Male	50	Direct microscopy (KOH), fungal culture (Mycosel Agar), PCR sequencing >(Calmodulin gene)	Pulmonary	*S. brasiliensis*	*S. brasiliensis*	[81]
		Pelotas	7	ND		Gram-stain microscopy, fungal culture (Sabouraud-dextrose agar added with chloramphenicol and Mycosel), PCR sequencing (ITS1 and ITS4 and Calmodulin gene)	Lymphocutaneous 4 (57.1%) Ocular 3 (42.9%)	*S. brasiliensis*	*S. brasiliensis*	[82]
		São Paulo	1	Female	12	Histopatology (Grocott stainin), fungal culture.	Immunoreactive cutaneous	*Sporothrix* spp.	*Sporothrix* spp.	[83]
		Recife	1	Male	25	Histopatology (hematoxylin–eosin straining), fungal culture (Sabouraud dextrose agar with chloramphenicol), PCR sequencing (using the species-specific primers Sbra-F and Sbra-R and Calmodulin gene)	Ocular	*S. brasilienis*	*S. brasilienis*	[84]
		Rio de Janeiro	1	Male	44	Fungal culture	Disseminated	*Sporothrix* spp.	*Sporothrix* spp.	[85]
		São Paulo	2	Male	3 and 12	Fungal culture	Ocular	*Sporothrix* spp.	*Sporothrix* spp.	[86]
		ND	1	Female	45	Histopathology, Fungal culture (Sabouraud dextrose agar), PCR sequencing (Whole genome sequencing)	Cutaneos carbuncle	*S. brasiliensis*	*S. brasiliensis*	[87]
		Rio de Janeiro	1	Male	11	Fungal culture (Sabouraud’s dextrose agar), Culture microscopy with Lactofenol blue	Facial Cutaneous	*Sporothrix* spp.	*Sporothrix* spp.	[88]
		Guarulhos, Sao Paulo	1	Male	56	Fungal culture, Histopathology (Peryodic Acid Schiff staining),	Disseminated	*Sporothrix* spp.	*Sporothrix* spp.	[89]
		São Paulo	1	Female	39	Fungal culture (Sabouraud agar)	Lymphocutaneous	*Sporothrix* spp.	*Sporothrix* spp.	[90]
		Brasilia	1	Male	26	Fungal culture	Disseminated	*Sporothrix* spp.	*Sporothrix* spp.	[91]
		Rio de Janeiro	4 from 2006–2016	Female Age ranged from 18–34	Average 25	Fungal culture, PCR sequencing (Primer T3B fingerprintig assay)	Fixed cutaneous 2 (50%) Lymphocutaneous 2 (50%)	*Sporothrix* spp. 2 (50%) *S. brasiliensis* 2 (50%)	*Sporothrix* spp. 2 (50%) *S. brasiliensis* 2 (50%)	[92]
		Rio de Janeiro	3 from 2006 to 2013	Male Age ranged from 25–43	Average 32	Fungal culture, PCR sequencing (primer T3B fingerprinting assay )	Disseminated 3	*S. brasiliensis*	*S. brasiliensis*	[93]
		Rio de Janeiro	1	Male	66	Direct microscopy, fungal culture (Sabouraud dextrose agar, potato dextrose agar, corn meal agar and and brain heart infusion agar), Histopatology, PCR sequencing (Calmodulin gene)	Lymphocutaneous	*S. globosa*	*S. globosa*	[94]
		Palmeira das Missões	1	Male	73	Fungal culture	ND	*S. schenckii* complex	*Sporothrix* spp.	[95]
		Rio de Janeiro	1	Male	5	Fungal culture and Histopatology	Osteoarticular	*S. schenckii*	*Sporothrix* spp.	[96]
		Rio de Janeiro	1	Male	61	Fungal culture, PCR sequencing (primer T3B fingerprinting assay)	Disseminated	*S. brasiliensis*	*S. brasiliensis*	[97]
		ND	1	Male	49	Fungal culture	Disseminated	*S. schenckii*	*Sporothrix* spp.	[98]
		Espírito Santo	3	Female	30 and 10	Direct microscopy (KOH), fungal culture (Sabouraud Dextrose agar and Mycosel agar^®^), assimilation of sugar test	Chancre 3	*S. brasiliensis*	*S. brasiliensis*	[99]
				Male	14					
		Rio de Janeiro	1	Female	9	PCR sequencing (calmodulina gene)	Dacryocystitis	*S. brasiliensis*	*S. brasiliensis*	[100]
		Rio de Janeiro	2	Female	22 and 27	Fungal culture, PCR sequencing (Calmodulina gene)	Fixed Cutaneous	*S. brasiliensis*	*S. brasiliensis*	[101]
		Rio de Janeiro	1	Male	6	Fungal culture, PCR sequencing	Invasive Sinusitis	*S. brasiliensis*	*S. brasiliensis*	[102]
		Rio de Janeiro	1	Male	56	Fungal culture, PCR sequencing	Meningitis, Lymphocutaneous	*S. brasiliensis*	*S. brasiliensis*	[103]
		São Paulo	20 from 2012–2020	Male 9 (45%)	Age ranged from 2–81 mean 32.2 ± 25.10	Fungal culture	ND	*Sporothrix* spp.	*Sporothrix* spp.	[104]
				Females 11 (55%)						
		Rio de Janeiro	64 from 2013–2015	ND		Fungal culture (Sabouraud Dextrosa Agar, Mycosel )	Lymphocutaneous 43 (67%) Fixed cutaneous 21 (33%)	*S. schenckii* *sensu lato*	*Sporothrix* spp.	[105]
		Minas Gerais 1 Ceará 1 Goiás 1 Pernambuco2 São Paulo 1	6	ND		Fungal culture (Potato Dextrose agar, Corn Meal agar), Carbohydrate assimilation tests, PCR sequiencing (calmodulin gene)	Lymphocutaneous 2 (33.3%) Disseminated 1 (16.7%) ND 3 (50%)	*Sporothrix mexicana* 3 (50%) *Sporothrix globosa* 3 (50%)	*Sporothrix mexicana* 3 (50%) *Sporothrix globosa* 3 (50%)	[106]
	Colombia	Antioquia	34	ND		Fungal culture, PCR sequencing (ITS 1–2 and β-tubulin)	ND	*S. schenckii* sensu stricto 22 (65.7%) *S. globosa* 12 (34.2%)	*S. schenckii* 22 (65.7%) *S. globosa* 12 (34.3%)	[56]
		Bogotá	2.28% (14 cases/612 patients)	Male ND Female ND	Between: 0–18	Fungal culture	ND	*Sporothrix* spp.	*Sporothrix* spp.	[107]
		Casanare	1	Male	18	Fungal culture, Histopathology	Verrucose	*Sporothrix* spp.	*Sporothrix* spp.	[108]
		Marandúa	1	Female	48	Fungal culture, Histopathology	Fixed cutaneous	*S. schenckii* sensu lato	*Sporothrix* spp.	[109]
	Chile	Santiago	1	Male	54	Histopathology	Lymphocutaneous	*Sporothrix* spp.	*Sporothrix* spp.	[110]
		Valparaíso	1	Female	75	Fungal culture Direct microscopy, Sugar assimilation (sucrose)	Lymphocutaneous	*Sporothrix globosa*	*Sporothrix globosa*	[111]
		Viña del Mar	1	Female	64	Direct microscopy, Fungal culture (Sabouraud with cycloheximide and potato dextrose agar) nitrogen-based agar, sequencing (D1/D2 region of the fungal 26S rRNA gene, it region; a partial fragment of the β-tubulin gene; ITS 1 and 2; and the 5.8S gene (SU)).	Onychomycosis	*Sporothrix pallida*	*Sporothrix pallida*	[112]
	Paraguay	Itá	2	Male Male	52	Histopathology (Peryodic Acid Schiff), Fungal culture, direct microscopy with Giensa strein	Lymphocutaneous 1 (50%) Fixed cutaneous 1 (50%)	*Sporothrix* spp.	*Sporothrix* spp.	[113]
		Cordillera 2 Guairá, Central 2 Misiones 2 San Pedro 2 Caaguazú 1	11 from 1997–2019.	Male 10 Female 1	Mean Age: 37,6 ± 20 Range: 24–69	Direct microscopy (KOH 10%), fungal culture (Sabouraud agar with glucose 2%, potato dextrose agar with chloramphenicol),	Lymphocutaneous 11 (100%)	*Sporothrix schenckii* complex	*Sporothrix* spp.	[114]
	Perú	Apurímac	2	Female	65	Direct microscopy, Giemsa stain Culture Microscopy with lactophenol cotton blue, Carbohydrate assimilation test (sucrose and raffinose) in nitrogen base	Fixed cutaneous	*S. schenckii*	*S. schenckii*	[115]
				Female	67					
		Apurímac	21	Male (12) Female (9)	Average: 9	Fungal culture	Lymphocutaneous 13 (62%) Fixed cutaneous 8 (38%)	*Sporothrix* spp.	*Sporothrix* spp.	[116]
		Apurímac 2850	57 (15/100,000)	Male 1734 Female 1255	ND	Fungal culture, Microscopy with lactophenol cotton blue and PCR sequencing	Lymphocutaneous 2942 (63%) Fixed cutaneous 1728 (37%)	*S. schenckii* 4651 (99.6%) *S. schenckii* sensu stricto 19 (0.4%)	*Sporothrix* spp. 4651 (99.6%) *S. schenckii* 19 (0.4%)	[117]
		Cajamarca 1500	30 (3/100,000)							
		La Libertad 100	4 (0.5/100,000)							
		Cusco 200	2 (0.2/100,000)							
		Otras regiones 20	≤1 (0.1/100,000)							
		Abancay	1	Male	6	Fungal culture	Lymphocutaneous	*Sporothrix* spp.	*Sporothrix* spp.	[118]
		Lima	1	Male	23	Fungal Culture Microscopy with lactophenol blue, MALDI-TOF MS, PCR sequencing (D1/D2 region of the fungal 26S rRNA gene)	Fixed cutaneous	*S. schenckii*	*S. schenckii*	[119]
		Lima	1	Male	42	Histopathology, Microscopy, Fungal Culture	Disseminated cutaneous	*S. schenkii* sensu lato	*Sporothrix* spp.	[120]
		Cajamarca	94 from 1991 to 2014	Males (67) Female (27)	Average: 36	Direct microscopy, Gram and Giemsa stain, Fungal culture, Histopathology	Lymphocutaneous 44 (47%) Fixed cutaneous 37 (39%) Disseminated cutaneous 11 (12%) Extra-cutaneous 1 (1%) ND 1 (1%)	*S. schenckii*	*Sporothrix* spp.	[121]
		Apurímac								
		Amazonas								
		Ancash	1	Male	58	Fungal culture	Lymphocutaneous	*S. schenckii*	*Sporothrix* spp.	[122]
		Cusco	1	Female	53	Fungal culture (Sabouraud)	Disseminated	*S. schenckii*	*Sporothrix* spp.	[123]
	Uruguay	Tacuarembó 10 Cerro Largo 9 Canelones 9 Montevideo 5 Rocha 4 Paysandú 3 Flores 3 Río Negro 2 Colonia 2 Artigas 1 Rivera 1 Maldonado 1 Soriano 1 Non-registered 20	157 from 1983 to 2020	Male (152)	13–79 age range	Gram staining and culture in Sabouraud	Nodular Lymphatic 120 (76.4%) Fixed cutaneous 30 (19.1%) ND 7 (4.5%)	*Sporothrix* spp.	*Sporothrix* spp.	[124]
				Female (5)						
	Venezuela	Caracas	68	ND		Fungal culture, PCR sequencing (Calmodulin locus and ITS regions)	ND	*S. schenckii* 42 (62%) *S. globosa* 26 (38%)	*S. schenckii* 42 (62%) *S. globosa* 26 (38%)	[125]
		Aragua 55 Miranda 32 Other states 46	133 from 1963–2019	Male (95) Female (38)	0–15 15–30 >30	Direct microscopy Fungal culture	Lymphocutaneous 84 (63.15%) Fixed cutaneous 48 (36.09%) Cornea 1 (0.7%)	*S. schenckii* sensu lato 130 (97.7%) ND 3 (2.3%)	*Sporothrix* spp.	[126]
		Bolívar 14	0.55% (220 cases/39,806 patients)	ND	25–45 years	Microscopy and fungal culture	ND	*Sporothrix* spp.	*Sporothrix* spp.	[127]
		Caracas 160								
		Carabobo 6								
		Falcón 3								
		Lara 5								
		Mérida 1								
		Monagas 24								
		Sucre 1								
		Táchira 2								
		Zulia 4								
		Costal Range 22	31 from 1973–2013	Male 64% Female 36%		Microscopy, fungal culture, pruebas bioquímicas, PCR sequencing (Calmodulin gene and ITS 4–5)	Fixed cutaneous 18 (60%) Lymphocutaneous 11 (36.33%) Disseminated 1 (3.33)%	*S. schenckii* sensu stricto 17 *S. globosa* 13 and *Ophiostoma stenoceras* 1	*S. schenckii* 17 (56.67%) *S. globosa* 13 (43.33%)	[128]
		Andes 7								
		Plains 2								

ND: Not Determined.

## Data Availability

Not applicable.

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
