# Peer review of "Epidemiology of Clinical Sporotrichosis in the Americas in the Last Ten Years"

_jof, 2022, doi:10.3390/jof8060588_

Round 1
Reviewer 1 Report
The authors the analyzed the human sporotrichosis cases reported in the Americas in the past ten years. The research is desired since the occurrence of sporotrichosis is unknown.
It is an interesting manuscript, which content brings new information. However, some points should be reviewed, as it follows:
Abstract
Lines 48-49
Besides the climate, other factors must be taken into account, besides the climate, like the zoonotic transmission in Brazil.
I think that “high occurrence of cases reported in the literature” would be more appropriate than “prevalence”.
Introduction
Lines 56-57
“Sporotrichosis is a fungal infection caused by species of the complex Sporothrix schenckii (S. schenckii sensu stricto, S. globosa, S. mexicana, S. luriei, S. albicans, S. brasiliensis).”
The denomination of a species complex is no longer appropriate. Sporotrichosis is a fungal infection caused by species of the Sporothrix genus. There is currently a suggestion to adopt the term ‘‘clinical clade’’ or ‘‘pathogenic clade’’ to refer to S. brasiliensis, S. schenckii, S. globosa and S. luriei. The remaining Sporothrix are nested in an ‘‘environmental clade’’. The environmental species are rare agents of infections in mammals and are distributed in different clades, such as the S. pallida complex (e.g., S.chilensis, S. mexicana, S. humicola and S. pallida and the S. stenoceras complex.
Rodrigues, A. M. et al. The threat of emerging and re-emerging pathogenic Sporothrix species. Mycopathologia, 2020.
Lines 67-69
“Another form of transmission, although sporadic, is reported to result from scratches, bites, pecks, and stingers of different animals, but it is still a rare and infrequent form of transmission [4,5].”
The zoonotic form of transmission related to cats due to S. brasiliensis is not rare. On the contrary, sporotrichosis transmission profile in Brazil is different from other countries, mainly due to the predominance of zoonotic transmission by domestic cats. Most of the Brazilian cases (79.3%) collected in a recent systemic review reported cat zoonotic transmission of sporotrichosis. Also, the occurrence of zoonotic sporotrichosis due to S. brasiliensis in Argentina and Paraguay points to a potential transboundary expansion of this species to other coutries in South America.
Gremião IDF et al (2020) Geographic expansion of sporotrichosis, Brazil. Emerg Infect Dis 26:621–624
Rabello VBS et al. The Historical Burden of Sporotrichosis in Brazil: a Systematic Review of Cases Reported from 1907 to 2020. Braz J Microbiol. 2022 Mar;53(1):231-244. doi: 10.1007/s42770-021-00658-1. Epub 2021 Nov 26. PMID: 34825345; PMCID: PMC8882507.
Material and Methods
In the Flow Diagram, it was not clear why 50 records were excluded. What were the exclusion criteria?
Discussion
Lines 209-213
Why the authors concluded that the highest number of reports in South America is due to its climate? Some points should be better discussed here. Different from some regions of South America, where the disease occurs more frequently during humid autumn and summer seasons, in Mexico, the incidence spikes during the cold and dry seasons. In Brazil, the high occurrence of sporotrichosis is principally related to the zoonotic route, in which felines participate actively in the disease transmission.
Table 1:
“Rio de Janeiro” instead of “Río de Janeiro”
“Espírito Santo” instead of “Espiritu Santo”
Author Response
We greatly appreciate the revisions made to our manuscript entitled "Epidemiology of Sporotrichosis in the Americas in the last ten years" by Arenas et al.
We believe that we have addressed all comments and suggestions from the reviewers.
These revisions have been of great help to round out and improve our work. We hope that the work will be suitable for acceptance and publication in your prestigious journal with these changes.
We have included the reviewer's comments immediately after this letter and indicated precisely how we addressed each concern or problem.
We thank you for the opportunity to correct this manuscript and your continued interest in our research. We look forward to hearing from you shortly.
Reviewer’s comments, author responses, and manuscript changes
We are thankful to the referees for carefully reviewing the manuscript and the opinions regarding its presentation. In what follows, the referee’s comments are in italics, the author's responses are in blue, and the changes made are highlighted.
Reviewer 1
The authors the analyzed the human sporotrichosis cases reported in the Americas in the past ten years. The research is desired since the occurrence of sporotrichosis is unknown.
Response: We are thankful for the time and effort you have invested in revising our manuscript. All your suggestions have enriched our work. Please find our answers to your valuable recommendations; we hope that we have addressed all your concerns.
It is an interesting manuscript, which content brings new information. However, some points should be reviewed, as it follows
- Abstract
Lines 48-49
Besides the climate, other factors must be taken into account, besides the climate, like the zoonotic transmission in Brazil.
I think that “high occurrence of cases reported in the literature” would be more appropriate than “prevalence”.
Response: This is an excellent observation. We have added information in the "Abstract” and corrected the paragraphs as follows:
Lines 49-53
Background: Sporotrichosis is a fungal infection caused by species of the Sporothrix genus. Presently, the prevalence of sporotrichosis in the Americas is unknown, so this study aims to analyze the cases reported in the past ten years. Methods: An advanced search was conducted from 2012 to 2022 in English and Spanish in PUBMED, SciELO, and Cochrane, with the terms: "sporotrichosis," "lymphocutaneous sporotrichosis," "fixed sporotrichosis," "mycosis," "Sporothrix spp.", "Sporothrix complex", "S. schenckii sensu stricto", "S. schenckii sensu lato”, “S. globose”, “S. brasiliensis”, “S. luriei”. Sporotrichosis is a fungal infection caused by species of the Sporothrix genus associated with "pathogenicity" or "epidemiology." Results: 124 articles were found in the Americas, corresponding to 12568 patients. 87.38% of cases were reported in South America, 11.62% in North America, and 1.00% in Central America and the Caribbean. Brazil, Peru, and Mexico had the highest number of cases. The most prevalent etiological agents were S. schenckii complex/ Sporothrix spp. (52.91%), S. schenckii (42.38%), others (4.68%) and Not Determined (ND) (0.03%). The most frequent form of the disease was the lymphocutaneous infection; however, the infection type was not determined in 5639 cases. Among the diagnostic methods, culture was the most used. Conclusions: There is a high occurrence of cases reported in the literature. South America is the region with the highest number of reports because of its environment (climate, inhalation of spores, etc.), zoonotic transmission (scratches and sneezes from contaminated animals) and outdoor activities with possible traumatic inoculation (agriculture, gardening, and related occupations). Molecular diagnosis has not been sufficiently developed due to its high cost.
- Lines 56-57
“Sporotrichosis is a fungal infection caused by species of the complex Sporothrix schenckii (S. schenckii sensu stricto, S. globosa, S. mexicana, S. luriei, S. albicans, S. brasiliensis).”
The denomination of a species complex is no longer appropriate. Sporotrichosis is a fungal infection caused by species of the Sporothrix genus. There is currently a suggestion to adopt the term ‘‘clinical clade’’ or ‘‘pathogenic clade’’ to refer to S. brasiliensis, S. schenckii, S. globosa and S. luriei. The remaining Sporothrix are nested in an ‘‘environmental clade’’. The environmental species are rare agents of infections in mammals and are distributed in different clades, such as the S. pallida complex (e.g., S.chilensis, S. mexicana, S. humicola and S. pallida and the S. stenoceras complex.
Rodrigues, A. M. et al. The threat of emerging and re-emerging pathogenic Sporothrix species. Mycopathologia, 2020.
Response: Thank you for your kind and appropriate observation. We have rewritten part of the "Section 1. Introduction" and corrected the paragraphs as follows:
Lines 60-67
Sporotrichosis is a fungal infection caused by thermo-dimorphic fungi species of Sporothrix genus. Nowadays, Sporothrix have been classified into clinical clade: Sporothrix schenckii, Sporothrix globosa, Sporothrix brasiliensis and Sporothrix luriei, although on some occasions the species of the environmental clade, Sporothrix pallida, Sporothrix mexicana and Sporothrix chilensis may cause infection upon contact with the individual. The infections occur mainly cutaneously or subcutaneously with lymphatic involvement [1-4]. This infection has been considered in Latin America as the most frequent subcutaneous mycosis [2].
- Lopes-Bezerra, L.M.; Mora-Montes, H.M.; Zhang, Y.; Nino-Vega, G.; Rodrigues, A.M.; de Camargo, Z.P.; de Hoog, S. Sporotrichosis between 1898 and 2017: The evolution of knowledge on a changeable disease and on emerging etiological agents. Medical Mycology 2018, 56, S126-S143.
- Rabello, V.B.S.; Almeida, M.A.; Bernardes-Engemann, A.R.; Almeida-Paes, R.; de Macedo, P.M.; Zancopé-Oliveira, R.M. The Historical Burden of Sporotrichosis in Brazil: a Systematic Review of Cases Reported from 1907 to 2020. Braz J Microbiol. 2022, 53, 231-244.
- Rodrigues, A.M.; Della Terra, P.P.; Gremião, I.D.; Pereira, S.A.; Orofino-Costa, R.; de Camargo, Z.P. The threat of emerging and re-emerging pathogenic Sporothrix Mycopathologia. 2020, 185, 813-842. doi: 10.1007/s11046-020-00425-0.
- Gremião, I.D.F.; Evangelista Oliveira, M.M.; Monteiro de Miranda, L.H.; Saraiva Freitas, D.F.; Peraira, S.A. Geographic Expansion of Sporotrichosis, Brazil . Emerg Infect Dis 2020, 26, 621-662.
- Lines 67-69
“Another form of transmission, although sporadic, is reported to result from scratches, bites, pecks, and stingers of different animals, but it is still a rare and infrequent form of transmission [4,5].”
The zoonotic form of transmission related to cats due to S. brasiliensis is not rare. On the contrary, sporotrichosis transmission profile in Brazil is different from other countries, mainly due to the predominance of zoonotic transmission by domestic cats. Most of the Brazilian cases (79.3%) collected in a recent systemic review reported cat zoonotic transmission of sporotrichosis. Also, the occurrence of zoonotic sporotrichosis due to S. brasiliensis in Argentina and Paraguay points to a potential transboundary expansion of this species to other coutries in South America.
Gremião IDF et al (2020) Geographic expansion of sporotrichosis, Brazil. Emerg Infect Dis 26:621–624
Rabello VBS et al. The Historical Burden of Sporotrichosis in Brazil: a Systematic Review of Cases Reported from 1907 to 2020. Braz J Microbiol. 2022 Mar;53(1):231-244. doi: 10.1007/s42770-021-00658-1. Epub 2021 Nov 26. PMID: 34825345; PMCID: PMC8882507
Response: We thank the reviewer for this observation. In this new version, we have changed the information about zoonotic form of transmission in "Section 1. Introduction" and extended the information in the "Section 6. Discussion"
"Section 1. Introduction"
Lines 75-78
Another form of transmission, which has been increasing in recent times in some regions of the continent such as Brazil, Argentina, Paraguay, and Panama, has been reported to result from scratches, bites, pecks, and stingers of different animals [1-4].
"Section 6. Discussion"
Lines 222-231
The latter could explain in some part why the high prevalence in Latin America being endemic in this region [1-4,129]. However, in these three countries (Brazil, Peru and Mexico) has specific characteristics that increases the number of cases. In Brazil, and adjacent countries (ex. Argentina, Paraguay) the increasing numbers of cases has been associated to zoonotic infection mainly from infected cats through scratches or sneezes [3,4]. Since the zoonotic transmission of S. brasiliensis is the most important form of communication, it is recommended that hygienic measures be taken regarding domestic animals such as cats, rodents, etc. due to possible infections related to these. If it is diagnosed in animals, it must be treated immediately, as well as the use of gloves when handling animals with injuries [2-4].
- Lopes-Bezerra, L.M.; Mora-Montes, H.M.; Zhang, Y.; Nino-Vega, G.; Rodrigues, A.M.; de Camargo, Z.P.; de Hoog, S. Sporotrichosis between 1898 and 2017: The evolution of knowledge on a changeable disease and on emerging etiological agents. Medical Mycology 2018, 56, S126-S143.
- Rabello, V.B.S.; Almeida, M.A.; Bernardes-Engemann, A.R.; Almeida-Paes, R.; de Macedo, P.M.; Zancopé-Oliveira, R.M. The Historical Burden of Sporotrichosis in Brazil: a Systematic Review of Cases Reported from 1907 to 2020. Braz J Microbiol. 2022, 53, 231-244.
- Rodrigues, A.M.; Della Terra, P.P.; Gremião, I.D.; Pereira, S.A.; Orofino-Costa, R.; de Camargo, Z.P. The threat of emerging and re-emerging pathogenic Sporothrix Mycopathologia. 2020, 185, 813-842. doi: 10.1007/s11046-020-00425-0.
- Gremião, I.D.F.; Evangelista Oliveira, M.M.; Monteiro de Miranda, L.H.; Saraiva Freitas, D.F.; Peraira, S.A. Geographic Expansion of Sporotrichosis, Brazil . Emerg Infect Dis 2020, 26, 621-662.
- Orofino-Costa, R.; Marques-de-Macedo, P.; Messias-Rodrigues, A.; Bernardes-Engemann, A.R. Sporotrichosis: an update on epidemiology, etiopathogenesis, laboratory and clinical therapeutics. An Bras Dermatol. 2017, 92, 606-620.
- Material and Methods
In the Flow Diagram, it was not clear why 50 records were excluded. What were the exclusion criteria?
Response: We thank the reviewer for this observation. In this new version, we have added information in the PRISMA diagram (Figure 2. Flowchart of the different phases of the systematic review) for the exclusion criteria in each phase of the process
Above: Line 101
Figure 2. Flowchart of the different phases of the systematic review
- Discussion
Lines 209-213
Why the authors concluded that the highest number of reports in South America is due to its climate? Some points should be better discussed here. Different from some regions of South America, where the disease occurs more frequently during humid autumn and summer seasons, in Mexico, the incidence spikes during the cold and dry seasons. In Brazil, the high occurrence of sporotrichosis is principally related to the zoonotic route, in which felines participate actively in the disease transmission.
Response: Thank you for the opportunity to clarify this point. We added information in “Section 6. Discussion” and corrected the paragraphs as follows:
Lines: 232-241
In Mexico, sporotrichosis is considered endemic and an occupational disease due to the different sources of infection. The climate of some regions in Mexico is perfect for the characteristics of this type of mycosis to increase its incidence. Although, tropical and subtropical are the climate preferred by this fungus, in this country, during the cold and dry seasons are the contagion peaks of these pathological agents. The states that are more affected are Mexico City, Puebla, Jalisco, Michoacan, State of Mexico and Guanajuato. In these states, the agriculture is one of the most important economic activities, so these most explain the high incidence of the Sporothrix contagion [130]. Thus, the principal recommendation in this region is the use of gloves or long-sleeved clothing when carrying out work activities where these species are endemic.
- Table 1:
“Rio de Janeiro” instead of “Río de Janeiro”
“Espírito Santo” instead of “Espiritu Santo”
Response: Thank you for the opportunity to clarify this point. We have corrected the grammatical errors in the Table 3. Epidemiology of Sporotrichosis in South America
Reviewer 2
- Because sporotrichosis in main implantation or subcutaneous fungal infection in the Americas, especially in Latin America, you last ten years revision is very well welcomed and appreciated.
Response: We are thankful for the time and effort you have invested in revising our manuscript. All your suggestions have enriched our work. Please find our answers to your valuable recommendations; we hope that we have addressed all your concerns.
- The search strategy you employed using the "PRISMA" systematic review, is not in compliance to the current Sporothrix taxonomy, so some publications may be not found in your literature review.
According to the most recent publications on the taxonomy of the genus Sporothrix, this fungus is now is not considered a monotypic taxon, i.e., formed by a single species, but a biodiverse genus comprehending several pathogenic and environmental species.
For a long time, the medically relevant Sporothrix were classified as a “species complex,” called the S. schenckii. Nevertheless, with the progress of knowledge of the biology of Sporothrix species, the denomination of a species complex is no longer appropriate.
There is currently a suggestion to adopt the term ‘‘clinical clade’’ or ‘‘pathogenic clade’’ to refer to S. brasiliensis, S. schenckii, S. globosa and S. luriei, which are often isolated from human and animal cases. The remaining Sporothrix are nested in an ‘‘environmental clade,’’ where they are often associated with substrates that vary from the soil and decomposing organic matter to insects and plants. Indeed, these environmental species are rare agents of infections in mammals and are distributed in different clades, such as the S. pallida complex (e.g., S. chilensis, S. mexicana, S. humicola and S. pallida), etc.
I encourage you to include the current taxonomic classification of Sporotrix spp, including the current denominations of S. schenckii, S. brasiliensis, S. globosa, etc.
For the current Sporothrix taxonomy, please refer to
1 - Rodrigues, A.M., Della Terra, P.P., Gremiao, I.D., Pereira, S.A., Orofino-Costa, R. and de Camargo, Z.P. (2020), “The threat of emerging and re-emerging pathogenic Sporothrix species”, Mycopathologia, pp. 1–20.
2 - Lopes-Bezerra, L.M., Mora-Montes, H.M., Zhang, Y., Nino-Vega, G., Rodrigues, A.M., de Camargo, Z.P. and de Hoog, S. (2018), “Sporotrichosis between 1898 and 2017: The evolution of knowledge on a changeable disease and on emerging etiological agents.”, Medical Mycology, Vol. 56 No. suppl_1, pp. S126–S143.
Response: Thank you for the opportunity to clarify this point. We have changed the information in "Section 1. Introduction" considering the taxonomical classification that includes the clinical and environmental clades instead of Sporothrix schenckii complex and corrected the paragraphs as follows:
"Section 1. Introduction"
Lines 60-67
Sporotrichosis is a fungal infection caused by thermo-dimorphic fungi species of Sporothrix genus. Nowadays, Sporothrix have been classified into clinical clade: Sporothrix schenckii, Sporothrix globosa, Sporothrix brasiliensis and Sporothrix luriei, although on some occasions the species of the environmental clade, Sporothrix pallida, Sporothrix mexicana and Sporothrix chilensis may cause infection upon contact with the individual. The infections occur mainly cutaneously or subcutaneously with lymphatic involvement [1-4]. This infection has been considered in latin america as the most frequent subcutaneous mycosis [2].
- Lopes-Bezerra, L.M.; Mora-Montes, H.M.; Zhang, Y.; Nino-Vega, G.; Rodrigues, A.M.; de Camargo, Z.P.; de Hoog, S. Sporotrichosis between 1898 and 2017: The evolution of knowledge on a changeable disease and on emerging etiological agents. Medical Mycology 2018, 56, S126-S143.
- Rabello, V.B.S.; Almeida, M.A.; Bernardes-Engemann, A.R.; Almeida-Paes, R.; de Macedo, P.M.; Zancopé-Oliveira, R.M. The Historical Burden of Sporotrichosis in Brazil: a Systematic Review of Cases Reported from 1907 to 2020. Braz J Microbiol. 2022, 53, 231-244.
- Rodrigues, A.M.; Della Terra, P.P.; Gremião, I.D.; Pereira, S.A.; Orofino-Costa, R.; de Camargo, Z.P. The threat of emerging and re-emerging pathogenic Sporothrix Mycopathologia. 2020, 185, 813-842. doi: 10.1007/s11046-020-00425-0.
- Gremião, I.D.F.; Evangelista Oliveira, M.M.; Monteiro de Miranda, L.H.; Saraiva Freitas, D.F.; Peraira, S.A. Geographic Expansion of Sporotrichosis, Brazil . Emerg Infect Dis 2020, 26, 621-662.
Also, we have added a paragraph of this information in the "Section 6. Discussion" explaining the results found in this systemic review of the last ten years and the paragraphs is written as follows:
"Section 6. Discussion"
Lines 242-246
Regarding the etiological agents responsible for the types of sporotrichosis, it is important to specify that they were referred to both in the table and in the text in the way they were named in the articles that were analyzed. Since most of them were written and published before the changes in taxonomical classification, they do not consider the clinical and environmental clades classification instead of Sporothrix schenckii complex.
Additionally, in the "Section 6. Discussion" we added information about Ophiostoma stenoceras and the paragraphs is written as follows:
"Section 6. Discussion"
Lines 255-261.
Likewise, within the systematic review, an Ophiostoma stenoceras appears, which in the taxonomic classification of Sporothrix represented its sexual state, in the year the report was made. Nevertheless, since 2016, Beer et al., concluded through phylogenetic analyzes that the genus Sporothrix was different from the genus Ophiostoma, that was before considered its sexual state. Officially, the sexual status of Sporothrix is not known, and this case of Ophiostoma stenoceras was included according to the regulations that governed the taxonomy before the divorce between the two genders occurred [132].

Reviewer 2 Report
Dear authors
Because sporotrichosis in main implantation or subcutaneous fungal infection in the Americas, especially in Latin America, you last ten years revision is very well welcomed and appreciated.
The search strategy you employed using the "PRISMA" systematic review, is not in compliance to the current Sporothrix taxonomy, so some publications may be not found in your literature review.
According to the most recent publications on the taxonomy of the genus Sporothrix, this fungus is now is not considered a monotypic taxon, i.e., formed by a single species, but a biodiverse genus comprehending several pathogenic and environmental species.
For a long time, the medically relevant Sporothrix were classified as a “species complex,” called the S. schenckii. Nevertheless, with the progress of knowledge of the biology of Sporothrix species, the denomination of a species complex is no longer appropriate.
There is currently a suggestion to adopt the term ‘‘clinical clade’’ or ‘‘pathogenic clade’’ to refer to S. brasiliensis, S. schenckii, S. globosa and S. luriei, which are often isolated from human and animal cases. The remaining Sporothrix are nested in an ‘‘environmental clade,’’ where they are often associated with substrates that vary from the soil and decomposing organic matter to insects and plants. Indeed, these environmental species are rare
agents of infections in mammals and are distributed in different clades, such as the S. pallida complex (e.g., S. chilensis, S. mexicana, S. humicola and S. pallida), etc.
I encourage you to include the current taxonomic classification of Sporotrix spp, including the current denominations of S. schenckii, S. brasiliensis, S. globosa, etc.
For the current Sporothrix taxonomy, please refer to
1 - Rodrigues, A.M., Della Terra, P.P., Gremiao, I.D., Pereira, S.A., Orofino-Costa, R. and de Camargo, Z.P. (2020), “The threat of emerging and re-emerging pathogenic Sporothrix species”, Mycopathologia, pp. 1–20.
2 - Lopes-Bezerra, L.M., Mora-Montes, H.M., Zhang, Y., Nino-Vega, G., Rodrigues, A.M., de Camargo, Z.P. and de Hoog, S. (2018), “Sporotrichosis between 1898 and 2017: The evolution of knowledge on a changeable disease and on emerging etiological agents.”, Medical Mycology, Vol. 56 No. suppl_1, pp. S126–S143.
Author Response

(The authors gave the same response as above.)

Round 2
Reviewer 2 Report
Dear Authors
The review study is extremely relevant in addressing the Epidemiology of Sporotrichosis in the Americas in the last ten years, however, for the data presented to represent an updated survey, a careful review of the current nomenclature is provided. As pointed during the first round
revision, there were several misconceptions regarding Sporothrix phylogeny in the first manuscript version. In the current manuscript version, an updated taxonomy of Sporothrix spp have bee updated, especially in the abstractm but not in the main text. For example, in the topic “Epidemiology of Sporotrichosis in North America (line 105), the old nomenclature is maintained, like “senso lato, senso stricto, etc”.
The name Sporothrix schenkii sensu latu and sensu stricto is not adopted anymore, precisely it was based on the previous nomenclature of S. schenckii as a species complex. Therefore, this older nomenclature should be avoided during the text. We suggested the studies of De Beer et al. (2016), doi: 10.1016/j.simyco.2016.07.001; Rodrigues et al (2020), doi: 10.1007/s11046-020-00425-0 and Queiroz-Telles et al. (2022), doi: 10.1007/s12281-022-00429-x for a better understanding about that.
The nomenclature "Sporothrix schenckii complex" is outdated as phylogenetic studies published in the last decade has shown that this group is a separate genus inside the Ophistomatales order, which comprises more than 50 phylogenetic different species. Moreover, it is discouraged the use of the term “complex” in medical mycology for a clade such as this that includes well-defined species causing different disease symptoms, and that differ from each other in routes of transmission, virulence, and antifungal susceptibility. I suggest you to revise the studies (De Beer et al., 2016, doi: 10.1016/j.simyco.2016.07.001; Chen et al. 2016, doi: 10.1016/j.funbio.2015.09.003 and Marimon et al., 2007, doi: 10.1128/JCM.00808-07). Therefore, we encourage the authors to rewrite the manuscript by removing all the "Sporothrix schenckii complex" citations. In addition, phylogenetic studies from the Sporothrix genus are addressing the species S. schenckii, S. brasiliensis, S. globosa and S. luriei as belonging to the Pathogenic clade inside the genus. Thus, instead of using Sporothrix complex you can use the term Pathogenic clade, this approach can be found at the studies of De Beer et al. (2016), doi: 10.1016/j.simyco.2016.07.001; Rodrigues et al (2020), doi: 10.1007/s11046-020-00425-0 and Queiroz-Telles et al. (2022), doi: 10.1007/s12281-022-00429-x.
I would suggest to classify sporotrichosis as “an implantation (subcutaneous) mycosis. In this fungal infection, usually deeper organic sites are involved.
During the last decades, the epizoonotic transmission of S. brasiliensis in Brazil, affects thousands of humans and felines and hundreds of dogs. The cat transmitted sporotrichosis has crossed the border reaching Argentina and Paraguay. Could you include a paragraph related to this outbreak?
Based on the comments and suggestions presented above, we encourage the authors to carry out a review and re-present this work, which will contribute to the epidemiological understanding of the disea
Author Response
Reviewer’s comments, author responses, and manuscript changes
We are thankful to the referees for carefully reviewing the manuscript and the opinions regarding its presentation. In what follows, the referee’s comments are in italics, the author's responses are in blue, and the changes made in the first round of revision are highlighted in yellow and the second one in blue.
Reviewer 2
- The review study is extremely relevant in addressing the Epidemiology of Sporotrichosis in the Americas in the last ten years, however, for the data presented to represent an updated survey, a careful review of the current nomenclature is provided. As pointed during the first round revision, there were several misconceptions regarding Sporothrix phylogeny in the first manuscript version. In the current manuscript version, an updated taxonomy of Sporothrix spp have been updated, especially in the abstract but not in the main text. For example, in the topic “Epidemiology of Sporotrichosis in North America (line 105), the old nomenclature is maintained, like “senso lato, senso stricto, etc”.
- The name Sporothrix schenkii sensu latu and sensu stricto is not adopted anymore, precisely it was based on the previous nomenclature of S. schenckii as a species complex. Therefore, this older nomenclature should be avoided during the text. We suggested the studies of De Beer et al. (2016), doi: 10.1016/j.simyco.2016.07.001; Rodrigues et al (2020), doi: 10.1007/s11046-020-00425-0 and Queiroz-Telles et al. (2022), doi: 10.1007/s12281-022-00429-x for a better understanding about that.
- The nomenclature "Sporothrix schenckii complex" is outdated as phylogenetic studies published in the last decade has shown that this group is a separate genus inside the Ophistomatales order, which comprises more than 50 phylogenetic different species. Moreover, it is discouraged the use of the term “complex” in medical mycology for a clade such as this that includes well-defined species causing different disease symptoms, and that differ from each other in routes of transmission, virulence, and antifungal susceptibility. I suggest you to revise the studies (De Beer et al., 2016, doi: 10.1016/j.simyco.2016.07.001; Chen et al. 2016, doi: 10.1016/j.funbio.2015.09.003 and Marimon et al., 2007, doi: 10.1128/JCM.00808-07). Therefore, we encourage the authors to rewrite the manuscript by removing all the "Sporothrix schenckii complex" citations. In addition, phylogenetic studies from the Sporothrix genus are addressing the species S. schenckii, S. brasiliensis, S. globosa and S. luriei as belonging to the Pathogenic clade inside the genus. Thus, instead of using Sporothrix complex you can use the term Pathogenic clade, this approach can be found at the studies of De Beer et al. (2016), doi: 10.1016/j.simyco.2016.07.001; Rodrigues et al (2020), doi: 10.1007/s11046-020-00425-0 and Queiroz-Telles et al. (2022), doi: 10.1007/s12281-022-00429-x.
Response: We are thankful for the time and effort you have invested in revising our manuscript. All your suggestions have enriched our work. Please find our answers to your valuable recommendations; we hope that we have addressed all your concerns.
We have added information about the new classification throughout the text. However, we consider necessary to mention both taxonomic classifications during the text because, in case the reader wishes to consult said information, it could be confusing if he does not know the changes that the Sporothrix genus has undergone in the classification during the last 5 years. It should be noted that highlighted in blue, are the changes in the tables and in the main text which were added in this revision.
All text
Also, we only mention, in the total cases by region of the continent, the sporotrichosis with the new taxonomical classification. Continuing with this idea, a specific analysis was carried out in the "Section 6. Discussion" part about the reported case of Ophiostoma stenoceras, in which reference is made to the divorce between the Sporothrix and such species.
"Section 6. Discussion"
Line 284-290
In addition, both classifications were added to the tables in order to facilitate their analysis and understanding by reviewers and potential readers.
Tables 1-3
- During the last decades, the epizoonotic transmission of S. brasiliensis in Brazil, affects thousands of humans and felines and hundreds of dogs. The cat transmitted sporotrichosis has crossed the border reaching Argentina and Paraguay. Could you include a paragraph related to this outbreak?
Response: Thank you for your kind and appropriate observation. We have changed the title of the article as follows: Epidemiology of Clinical Sporotrichosis in the Americas in the last ten years. The title of the article was modified, adding the word clinic, this to refer to the fact that the cases are of human clinical origin and not of other animal species.
Also, we have rewritten part of the "Section 1. Introduction" and "Section 6. Discussion" and corrected the paragraphs as follows:
"Section 1. Introduction"
Lines 80-82
Another form of transmission, which has been increasing in recent times in some regions of the continent such as Brazil, Argentina, Paraguay, and Panama
“Section 6. Discussion
Lines: 284-290
In Brazil, and adjacent countries (ex. Argentina, Paraguay) the increasing numbers of cases has been associated to zoonotic infection mainly from infected cats through scratches or sneezes [3,4]. Since the zoonotic transmission of S. brasiliensis is the most important form of communication, it is recommended that hygienic measures be taken regarding domestic animals such as cats, rodents, etc. due to possible infections related to these. If it is diagnosed in animals, it must be treated immediately, as well as the use of gloves when handling animals with injuries [2-4].
- Based on the comments and suggestions presented above, we encourage the authors to carry out a review and re-present this work, which will contribute to the epidemiological understanding of the disease.
Response: We are thankful for the time and effort you have invested in revising our manuscript. All your suggestions have enriched our work.
